# Resident Satisfaction of Urban Green Spaces through the Lens of Landsenses Ecology

**DOI:** 10.3390/ijerph192215242

**Published:** 2022-11-18

**Authors:** Sinan He, Dingkai Chen, Xiaoqi Shang, Linwei Han, Longyu Shi

**Affiliations:** 1Key Laboratory of Urban Environment and Health, Key Laboratory of Urban Metabolism of Xiamen, Institute of Urban Environment, Chinese Academy of Sciences, Xiamen 361021, China; 2University of Chinese Academy of Sciences, Beijing 100049, China

**Keywords:** residential area, landsenses ecology, subjective senses, greenspace characteristics

## Abstract

Residents’ satisfaction of urban green space has been widely detected in living environments around the world. Most previous reports were performed with objective indicators to reflect the characteristics of vegetation and landscapes of residential green space. However, subjective senses as impact factors in the evaluation of residents’ satisfaction based on landsenses ecology are scarce. To address this, in this study, physical perception, aesthetic cognition, and psychological cognition as latent variables in a structural equation model were investigated to determine the residents’ satisfaction in Xiamen, in southeast China, a famously high green space coverage region. The results indicate that physical perception is the fundamental condition to improve residents’ satisfaction, while aesthetic cognition and psychological cognition are the direct factors that influenced residents’ satisfaction. Residents exhibit a preference for the residential green space which contains more biodiversity and landscape diversity, a higher biomass, and greater openness. In addition, the residents’ perception significantly related to greenspace characteristics. The results provide a scientific basis for urban green space planning and optimization of ecological resources’ allocation.

## 1. Introduction

Continuous fast urbanization and socio-environmental changes are positing challenges appropriating human population densification and a significant loss of green spaces [1,2]. As a critical part of the metropolitan biological system, urban green spaces (UGS) can improve the quality of the urban environment, beautify urban landscape, maintain the balance of the urban ecosystem, promote sustainable urban development, and enhance human well-being [3,4]. Expansion and planning of green space are of more concern, and have been widely detected in developed, developing countries, and slums around the world [5,6,7]. Nevertheless, residents and ecological environment are the main stakeholders of the city. Therefore, residents’ satisfaction degree of UGS could result in maintaining UGS or support urban greening projects [8,9,10].

To date, the common theoretical basis of the current evaluation in a number of studies includes satisfaction theory, environmental psychology, and human settlements theory, etc. [11,12]. Satisfaction theory suggests that both green space quality and quantity are in correlation with residents’ satisfaction and happiness [13,14,15]. These theories are evaluated from those aspects of aesthetic cognition, auditory perception, and residents’ perceptions [16,17]. Most previous studies on residents’ satisfaction have been performed in physical properties of green space. However, reports on various types and levels of the subjective human senses as impact factors in the evaluation of residents’ satisfaction of green space are scarce. For an expression of the residents’ satisfaction of green space, to investigate their physical properties only could be insufficient; it is necessary to take account of the residents’ satisfaction of green space from different aspects of the human perception of the landscape environment.

This study aims to (1) evaluate what types of perceptions and cognition for residential green space would impact residents’ general satisfaction and how they impacted it; (2) measure the differences between the subjective senses and objective indicators reflecting the characteristics of vegetation and landscapes of residential green space; and (3) investigate the correlation between residents’ subjective perceptions and green space characteristics. The result could contribute to identifying the critical factors that impact residents’ perceptions and satisfaction, seeking out the aspects that need to improve in urban residential green space planning and construction, and providing a scientific basis for urban, community, and residential green space planning and design.

## 2. Literature Review

### Satisfaction with UGS Based on Landsenses Ecology

In the 1960s, satisfaction theory had been applied to ecological and environmental quality evaluations. The community satisfaction scale was defined as residents’ subjective perception to environmental quality. Satisfaction with the environment is formed through a person’s individual characteristics (e.g., demographics), as well as their perceptions of biophysical properties of their surrounding environment [18]. Satisfaction consists of various perceptions, for instance, physical perception, perception of functionality, psychological cognition, aesthetic cognition, social cognition, and landscape cognition [19,20]. Masterson found that residents express place attachment, and a large percentage express some level of community attachment [21]. Tian explored how college satisfaction, sense of achievement, and student happiness contribute to freshmen’ sense of belonging [22].

In practice, people give or integrate one or some of their visions into a carrier through appropriate forms of expression, so that others (including themselves) can comprehend this or these visions from the carrier and its related forms of expression. These visions can guide or regulate what people say and do to promote and ensure sustainable development. These carriers are cities, neighborhoods or buildings, as well as paintings, calligraphy works, poems, novels, songs, or logos. We call the carrier with this property landsense. Landsenses ecology pays attention to the relationship between the ecological environment and people, the needs of residents. Landsenses ecology is a scientific discipline which put forward the concept of human perception of landscape environment, is based on ecological principles, and aims at sustainable development. Consequently, the theory of landsenses ecology can provide a theoretical framework for solving the above problems. It studies land-use planning, construction, and management from the natural elements, physical perceptions, psychological perceptions, socio-economic perspectives, process and risk, and other related aspects [23]. The natural elements include light, temperature, water, soil, wind, etc. Physical perceptions consist of sight, smell, audition, taste, and touch (e.g., senses of wind speed, wind direction, temperature, humidity, etc.). Psychological perceptions include some elements of religion, culture, vision, metaphor, security, community relations, well-being, etc. The different selections and combinations of these elements will have different land-use effects [23,24]. Therefore, endowing the objects with the fundamental elements of perceptions through urban planning, construction, and management will mean that residents perceiving them will thus obtain higher satisfaction. Landsenses ecology conducts research from residents’ subjective perception and demand, and provides a theoretical framework for the study of residents’ perception of the ecological environment.

Residents’ satisfaction based on landsenses ecology includes the perspectives of physical perception, aesthetical cognition, psychological cognition, and perception about public facilities. Figure 1 shows a conceptual model reflecting the relationships between the four types of perception and cognition. According to Maslow’s Hierarchy of Needs, physical needs are the fundamental needs for human beings. Human beings will pursue more higher-level needs such as psychological needs only if their fundamental physical needs are satisfied. Senses and aesthetic cognition are closely related. Thomas Aquinas held that the fundamental basis of aesthetic cognition is the subject’s empathy with the object, and the senses related to aesthetic cognition most closely were vision and audition. In addition, according to the isomorphism principle in Gestalt psychology, the reason that aesthetic experiences arise from the perception of human beings is because urban function elements and emotion contents are structurally similar to some physical perceptions of human beings [25]. To analyze how physical perception influences residents’ satisfaction and to investigate the relationship between physical perception, aesthetic cognition, and psychological cognition, hypotheses were formulated as follows:

**H1.** 
*Physical perception has a significantly positive impact on aesthetic cognition;*


**H2.** 
*Physical perception has a significantly positive impact on psychological cognition;*


**H3.** 
*Physical perception has a significantly positive impact on residents’ satisfaction.*


Aesthetic cognition is an ability to generate joyful spirits when aesthetic senses—mainly vision and audition—are stimulated by aesthetic objects. There are two levels of aesthetic cognition: sensation and perception. The former is a reflection of a specific feature of an object, while the latter is a completely emotional reconstruction of aesthetic objects through adding the aesthete’s belief, memory, habit, and other life experience on the former [25]. In this paper, aesthetic cognition refers to the former since only the impact of residential green space features to residents’ aesthetic reflection was considered. Nowadays, residential green space planners’ attention is progressively focused on landscape construction in addition to plant coverage. A positive landscape aesthetic experience can arouse an aesthetic evaluation of the object from a resident and let him/her generate affection, attachment, and other emotions to the landscape, thus causing a more intimate interaction between the resident and the landscape. To analyze the impact of aesthetic cognition on residents’ satisfaction and its relationship with psychological cognition, hypotheses were formulated as follows:

**H4.** 
*Aesthetic cognition has a significantly positive impact on psychological cognition;*


**H5.** 
*Aesthetic cognition has a significantly positive impact on residents’ satisfaction.*


Psychological cognition refers to the psychological experience of the environment from human beings in the interaction between human beings and the environment [26]. To date, demands in urban lives have been transferred from a material level to a spiritual level. As the subject of the residential environment, human beings are the primary factor which should be considered in environment construction. Existing research suggests that UGS has an ability to facilitate the control of negative emotions and to improve communication between residents, thus to promote residents’ satisfaction [27]. To analyze the impact of psychological cognition on residents’ satisfaction, a hypothesis was formulated as follows:

**H6.** 
*Psychological cognition has a significantly positive impact on resident’s satisfaction.*


According to the concept and research framework of the habitat environment science, a human settlement is the integration of five systems including nature, humans, society, residences, and a supporting network. In neighborhoods, except green space, other residential elements such as public facilities [28,29], location and transport [30,31], community services and security [32], house quality [33], and social and economic environments [34,35] will all influence residents’ satisfaction. To analyze how residents’ perception of these elements influences their satisfaction, hypotheses were formulated as follows:

**H7.** 
*Perception about public facilities has a significant regulating effect on the relationship between physical perception and psychological cognition;*


**H8.** 
*Perception about public facilities has a significant regulating effect on the relationship between aesthetic cognition and psychological cognition.*


Satisfaction theory has been employed in some research of residential green space and it turns out that both green space quantity and quality are in correlation with residents’ satisfaction and happiness [15]. The relationship between neighborhood green spaces and residential satisfactions considering both the quantity and quality of green space was investigated in the previous study [16]. Green space quantity is measured by the ratio of green space, and quality consists of plant community structure and foliar habits. Evaluation indicators include not only green space coverage, but also landscape aesthetics [36,37]. The accessibility and availability of green space were significantly correlated to residents’ perceptions of UGS quality and their satisfaction; moreover, the positive impact of accessibility and availability on the satisfaction was regulated by the residents’ subjective perceptions [15]. These works indicate that the residents’ perceptions have mediating effects on the relationship between objective indicators of residential green space and residents’ satisfaction. According to the theory of landsenses ecology, human perceptions of the environment come in a variety of types and levels [23]. How these different types and levels of perceptions affect and transmit between each other, and thus ultimately determine human satisfaction with the environment, requires further study.

## 3. Methods

### 3.1. Study Area

Residential green space from 6 different neighborhoods in Jimei District of Xiamen City were used as an example. Xiamen (24°23′–24°54′ N, 117°53′–118°26′ E) is an important central city of the south-east China coastal region, a harbor, and a famous tourist destination. The city consists of Xiamen Island, Gulang Island, some other small islands, and its mainland part. It covers a terrestrial area of 1699 km^2^ and an ocean area of 390 km^2^. Xiamen is located in a subtropical marine monsoon climate zone. It has a high percentage of forest coverage and holds beautiful natural landscapes. Thus, neighborhoods in Xiamen chosen for the study can be representative examples. Since green space is the most frequently used land type and residential green space is closely related to residents’ daily lives, i.e., the constructing quality of residential green space can impact the residents’ life quality and mental health [38] (Gascon et al., 2018), residential green space as a research object was embraced. To control variables such as economy, climate, and transport, six neighborhoods from Jimei District of Xiamen which have similar ages and building qualities were favored (Figure 2, Table 1).

### 3.2. Questionnaire and Data Collection

A questionnaire was drawn up to collect data for residents’ satisfaction analysis. Items were designed based on the eighteen observed variables in Table 2, in addition to the variable of residents’ satisfaction. Options of each answer were described as “very poor, poor, average, good, very good” according to the Likert five-point scale. The questionnaire consisted of three parts: information description, questions, and basic information. In the information description part, the concept of ecological infrastructure was introduced to respondents. The question part consisted of items about the nineteen variables. The last part was some questions about demographic characteristics. In February 2019, twenty-three people who were not the residents of the case neighborhoods were preliminarily examined. The results passed the test. Finally, 399 residents of the case neighborhoods were inspected in July 2019, which means the effective rate of the scrutinization was 89.26%. The data of demographic characteristics are shown in Appendix A, which indicates that the samples are representative of demographic characteristics.

The types and amounts of vegetation were derived from field inquiry. Landscape information such as types and colors were also derived from field inquiry. ArcGIS was adopted to obtain and calculate spatial data such as building area, height, volume, etc.

### 3.3. Data Analysis

#### 3.3.1. Analysis of Factors Influencing Residents’ Satisfaction

A structural equation model (SEM), which is extensively applied in the satisfaction evaluation domain, is a method that creates, evaluates, and verifies causal relationships based on a regression model. The impact mechanism of residential green space to residents’ satisfaction was investigated from the perspectives of physical perception, aesthetical cognition, psychological cognition, and perception about public facilities, taking residential green space from six different neighborhoods in Jimei District of Xiamen City as an example. As shown in Figure 2, physical perception (F1), aesthetic cognition (F2), psychological cognition (F3), and perception about public facilities (F4) were latent variables of the SEM. Table 2 shows latent variables and observed variables of SEM. The selection of observed variables is available in the Appendix A. Obtained data were processed by doing exploratory factor analysis (EFA), confirmatory factor analysis (CFA), reliability analysis, and validity analysis in sequential order. The processed data were then used in SEM validation and analysis to identify which latent variables had significant impact on residents’ satisfaction and how they influenced each other, and to identify which observed variables were of importance to the perceptions.

Quantifiable indicators as observed variables were selected for these latent variables. Observed variables of physical perception were selected in the aspects of vision, smell, audition, and touch. Taste was excluded because the ecological infrastructures in our case study mainly constituted of ornamental plants. Observed variables of aesthetic cognition were mainly selected from landscape aesthetic indicators. According to Tveit et al. (2006) [39], Wu et al. (2017) [40], Hur et al. (2010) [41], and Frank et al. (2013) [42], considering the features of urban residential ecological infrastructures, landscape richness, spatial conformity, management status, and green coverage were selected as the observed variables of aesthetic cognition. The management status refers to the sense of order and the vestige of manual maintenance. Excessive management can induce a feeling of manual work, while insufficient management gives a feeling of dilapidation [43]. Observed variables of psychological cognition include safety, belonging, esteem, and willingness to communicate based on Maslow’s Hierarchy of Needs.

#### 3.3.2. Measurement of Greenspace Characteristics

Indicators were picked to reflect the diversity, biomass, and openness of green space which, respectively, corresponded to landscape richness (I7), green coverage (I10), and visual field (I1) in observed variables [44,45]. The characteristics of residential green space vegetation and landscapes, their characterizing indicators, and their corresponding observed variables of perceptions are shown in Table 3.

Shannon–Wiener Index was used to characterize the biological diversity of green space. Urban tree leaf area regression model was used to evaluate the biomass of vegetation [46]. Building density and building space crowdedness was used to characterize landscape openness.

#### 3.3.3. Comparison of Subjective Senses and the Objective Indicator

The subjective perceptions of residents were compared with objective factors of vegetation and landscape characteristics of the residential green space. Each objective characteristic of vegetation and landscapes was measured using the average value of the normalized values of its characterizing indicators. Since building density and building space crowdedness were both negative indicators of landscape openness, a negative value of the result to measure landscape openness was implemented.

### 3.4. Data Quality Assurance

For quality assurance and control, the significant difference analysis, factor analysis, and confirmatory factor analysis were used.

In the significant difference analysis of influencing factors, the Kruskal–Wallis test instead of the one-way ANOVA was adopted. The results showed that there were significant differences in observed variables, except for temperature amenity (I2) and housing quality (I15), as shown in Appendix A. The average score of observed variables for each latent variable were compared among neighborhoods. The values of aesthetic cognition and psychological cognition are higher than the value of physical perception in case neighborhood C5, while in other case neighborhoods the value of physical perception is higher than the other two values (Appendix A).

In the exploratory factor analysis, Kaiser-Meyer-Olkin (KMO) and Bartlett’s test were performed on the rationality of the conceptual model. The KMO value was 0.903, and the *p*-value from Bartlett’s Test of Sphericity was lower than 0.001. The factor loadings of visual field (I1), plant odor (I5), and noise (I6) below 0.5 were excluded. Appendix A showed that the four common factors were identified, which are consistent with the four latent variables in the conceptual model (Figure 2). The cumulative variance value of more than 60% indicates that the respondents’ satisfaction with residential green space has been well explained. All factor loadings of the rotated component matrix value more than 0.58 indicates a strong correlation between the eighteen observed variables and the common factors.

In the confirmatory factor analysis, the composite reliability (CR) and average variance extracted (AVE) were computed to assess the convergent validity and discriminant validity. Appendix A shows the strong internal consistency among the latent variables of the model. In addition, the square roots of AVE are all greater than the inter-construct correlations.

## 4. Results

### 4.1. Influencing Factors of Residents’ Satisfaction

According to the conceptual model in Figure 2, there are five variables—physical perception (F1), aesthetic cognition (F2), psychological cognition (F3), perception about public facilities (F4), and residents’ satisfaction (I19)—constructing a model with eight paths. F1 and F4 were independent variables, while F2, F3 and I19 were dependent variables. The SEM shows a sufficient goodness-of-fit to the data: χ^2^/df = 1.67 < 3.00; TLI = 0.97 > 0.90; CFI = 0.98 > 0.90; RMSEA = 0.04 < 0.08; SRMR = 0.04 < 0.10. All goodness-of-fit results are within standard value ranges. Appendix A shows that all the path coefficients except for F1 → I19 and F1F4 → F3 are significant. Moreover, the R2-values of latent variables F2 and F3 are both greater than 0.3, indicating that these variables are explained in a high degree. Therefore, all hypotheses are supported, except H3 and H7. The result rejects hypothesis H7, which assumed that perception about public facilities (F4) has a regulating effect on the impact of physical perception (F1) upon psychological cognition (F3). As for hypothesis H3, which assumed that physical perception (F1) has a significant positive influence on residents’ satisfaction (I19), since the result shows that there is not direct impact of physical perception (F1) on residents’ satisfaction (I19), hypothesis H3 can only be supported when there is an indirect impact via perceptions about public facilities.

The indirect effects in paths ‘F1 → F2 → I19’, ‘F1 → F2F4 → F3 → I19’ and ‘F1 → F3 → I19’ were tested by products of coefficients. Appendix A shows that all the indirect effects in these three paths are significant, which indicates that physical perception (F1) has an indirect effect on residents’ satisfaction (I19) through three paths. Therefore, hypothesis H3 is also supported.

The path diagram of the structural equation model of residents’ satisfaction is shown in Figure 3. In summary, physical perception (F1), aesthetic cognition (F2), psychological cognition (F3), and perception about public facilities (F4) can affect residents’ satisfaction (I19) in residential green space. Among them, physical perception (F1) was the basic factor, which affected residents’ satisfaction (I19) by influencing residents’ aesthetic cognition (F2) and psychological cognition (F3). Aesthetic cognition (F2) and psychological cognition (F3) were the direct influencing factors of residents’ satisfaction (I19), and the influence of psychological cognition (F3) on residents’ satisfaction (I19) was slightly higher than that of aesthetic cognition (F4). In addition, residents’ perception of other facilities in the community (F4) can adjust the impact of residents’ aesthetic cognition (F2) on psychological cognition (F3).

### 4.2. Characteristics of Vegetation and Landscapes

Figure 4 shows the plant diversity in the six neighborhoods. The Shannon–Wiener species diversity indices of the six neighborhoods were relatively close. This indicates that the plant uniformity difference is small, and the species richness index is large in the six neighborhoods. Figure 5 shows the total biomass of vegetation in the six case neighborhoods. The C4 neighborhood had the highest biomass, reaching 278,000 m^2^, while the biomass in the C1 neighborhood was dramatically low, which was only 6346 m^2^. In the composition structure of the total biomass, herbs accounted for the largest proportion, followed by shrubs.

Figure 6 shows the colors and landscape types. Case neighborhoods C4 and C5 have a greater number of colors and landscape types than the other case neighborhoods, while C1 and C2 have the lowest number of colors, and C1 is the case neighborhood that has the lowest number of landscape types. The complexity of landscape components characterizing landscape diversity is also shown in Figure 6. C1 has the lowest complexity of landscape components, while both C4 and C5 have much higher complexity than the other case neighborhoods, which is approximately six to seven times that of C1. Figure 7 shows the building density and crowdedness of building space. Both building density and building space crowdedness of C3, C4, and C6 are relatively high, while they are relatively low in C2 and C5, indicating a better condition of landscape openness in C2 and C5.

### 4.3. Relationship between Residents’ Perception and Greenspace Characteristics

Table 4 shows the comparison result between the subjective perceptions and the objective characteristics of vegetation and landscapes in the form of standardized values. The distributions of the three objective characteristics across the six case neighborhoods are basically similar to the distributions of their corresponding subjective perceptions. The changes of subjective perceptions between case C2, C3, C4 and C5 can adequately reflect the changes of their corresponding objective characteristics. Case neighborhood C5 shows the greatest values of both objective characteristics and subjective perceptions in all the three comparison sub-figures. Case neighborhood C1 has the lowest values of both perceptions of landscape richness (I7) and green coverage (I10), and their corresponding objective characteristics. Case neighborhood C6 shows the lowest value of perception of visual field (I1), which is far from the values of other case neighborhoods, while the corresponding objective characteristic—the landscape openness—shows the lowest value in case C4.

## 5. Discussion

This study firstly constructed the structural equation model (SEM) to appropriately investigate the residents’ satisfaction. It then evaluated different types of perceptions and cognitions. Furthermore, the differences between the subjective senses and objective indicators were measured. At last, the correlation between residents’ subjective perceptions and green space characteristics was investigated. As such, this paper firstly applies the research of landsenses ecology to the residents’ satisfaction of urban green space, and also contains subjective perception and the objective environment, which enriches the structure of the model and seeks out the aspects that need to improve in urban residential green space planning and construction.

In terms of the evaluation of residents’ satisfaction of urban green space, aesthetic cognition and psychological cognition are directly related to residents’ satisfaction. Psychological perception exerts no direct effect on residents’ satisfaction, but the relationship between psychological perception and residents’ satisfaction can be formed through aesthetic cognition and psychological cognition. Similar to the study in Ningbo City in China, satisfaction of aesthetics and psychology had a higher impact than body satisfaction on the overall recreation satisfaction of urban residents [47]. This indicates that psychological cognition has a higher impact. The aesthetic cognition can not only influence resident satisfaction directly, but can also impact psychological cognition.

As for the perceptions that affect the residents’ satisfaction, the results show that perception to facilities can regulate the impact of aesthetic cognition on psychological cognition, which is consistent with the existing studies [45]. When residents have a low satisfaction level, the physical perception shows a higher indirect impact to psychological cognition through aesthetic cognition, and vice versa. Based on our research, it turns out that when residents have a lower perception level of residential facilities, their aesthetic cognition should be enhanced; when residents have a higher perception level of residential facilities, improving their physical perception can directly improve their psychological cognition, and thus improve their satisfaction.

In terms of landscape characteristics, the results show that residents show a preference for urban green space with higher diversity, biomass, and landscape openness. The lowest levels of diversity and biomass lead to the lowest levels of perceptions of landscape richness and green coverage, respectively. However, a low level of landscape openness does not guarantee a low level of visual field perception. There are still some other factors that have significant impact on the perception of the visual field. Similarly, in Hangzhou, the residents’ preferences for landscape elements and attributes of urban green space are examined by using principal components analysis [48].

Traditional urban planning, which is economy-orient, is over-reliant on urban comprehensive planning aiming at construction [49]. Thus, the green space system plays a subordinate role in traditional urban planning [50]. It is more in line with the people-oriented development model to explore the landscape elements and the combination the residents prefer, adopt a layout with affluent landscape types, a moderate natural level, and high resident satisfaction of facilities and services in urban green space system planning, and incorporate participants’ feelings and perceptions into the planning. In addition to urban ecological infrastructure construction, enhancing residents’ awareness of the importance of ecosystem services and improving their aesthetic cognition and experience through propagation, public education, and human–nature interaction can also help to increase the level of residents’ satisfaction.

This study has limitations that research on the relationship between landscape characteristics and residents’ perception can be further studied. In addition, study on the health factors of residents could be added.

## 6. Conclusions

Residents’ satisfaction of variant perspectives including physical perception, aesthetic cognition, psychological cognition, and perception about public facilities were investigated through a case study in six neighborhoods of Xiamen City, which have different green rates. The subjective perceptions were then compared with objective indicators reflecting the characteristics of residential green space vegetation and landscapes. The results show that a high level of physical perception is the fundamental condition to improve residents’ satisfaction. Aesthetic cognition and psychological cognition are the direct factors impacting residents’ satisfaction. Residents show a preference for residential green space with a higher species and landscape diversity, a higher biomass, and greater openness. The result can provide a scientific foundation for planners to identify the critical factors impacting residents’ perceptions and satisfaction, and to further improve the planning and design of urban green space.

## Figures and Tables

**Figure 1 ijerph-19-15242-f001:**
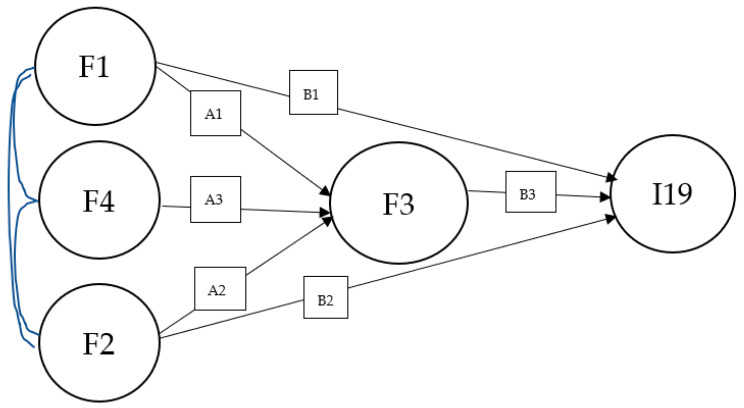
Structural equation model (SEM) meta-model.2.2. Linking Satisfaction and UGS Characteristics.

**Figure 2 ijerph-19-15242-f002:**
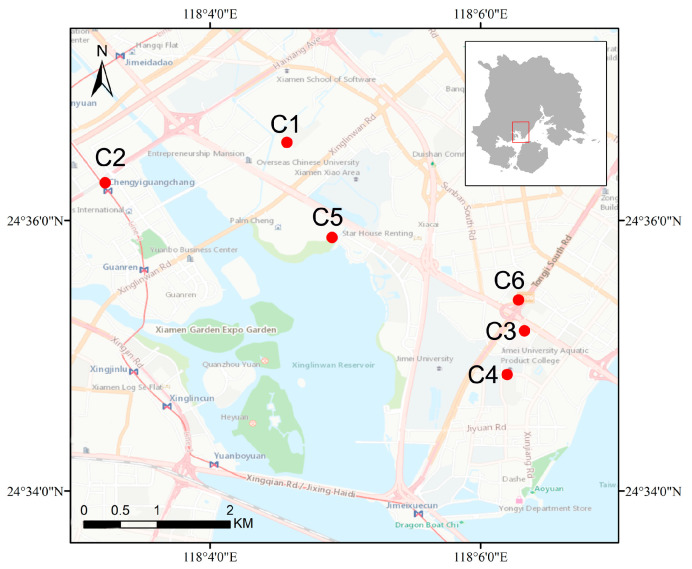
The study area and the case neighborhoods.

**Figure 3 ijerph-19-15242-f003:**
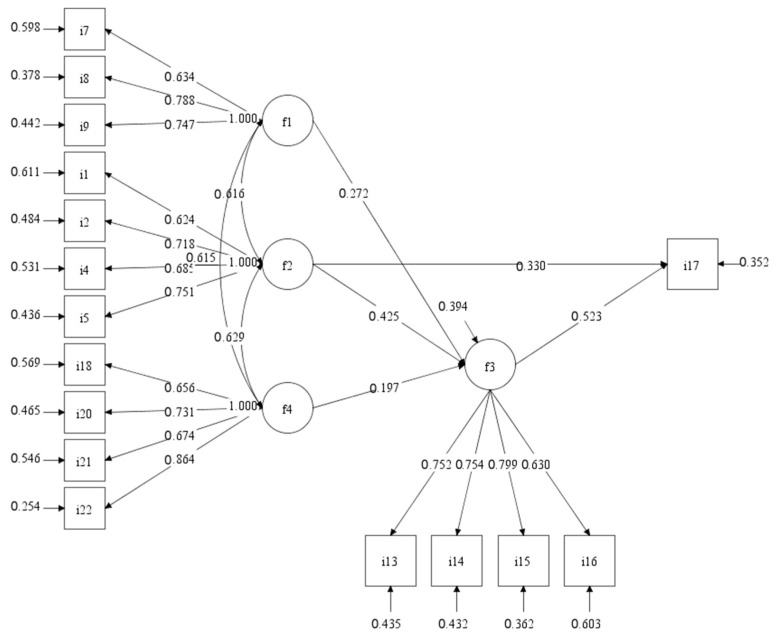
Path diagram of the structural equation model of residents’ satisfaction.

**Figure 4 ijerph-19-15242-f004:**
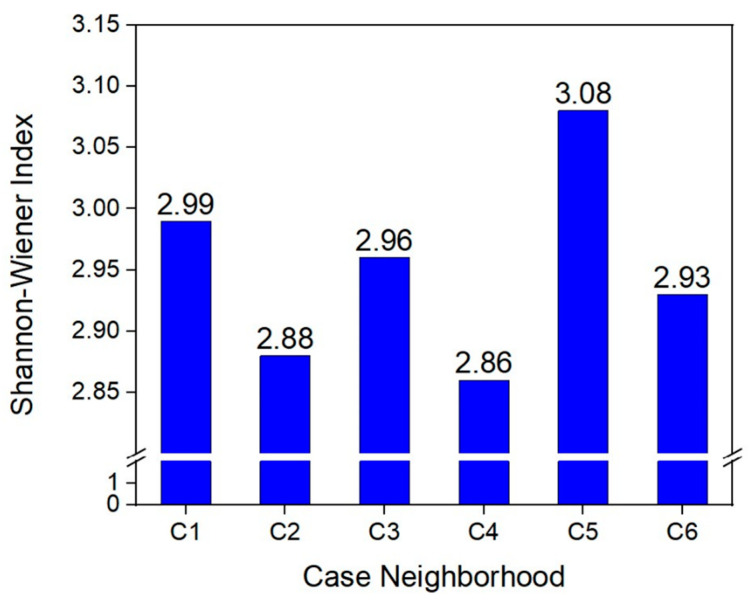
The plant diversity in the six neighborhoods.

**Figure 5 ijerph-19-15242-f005:**
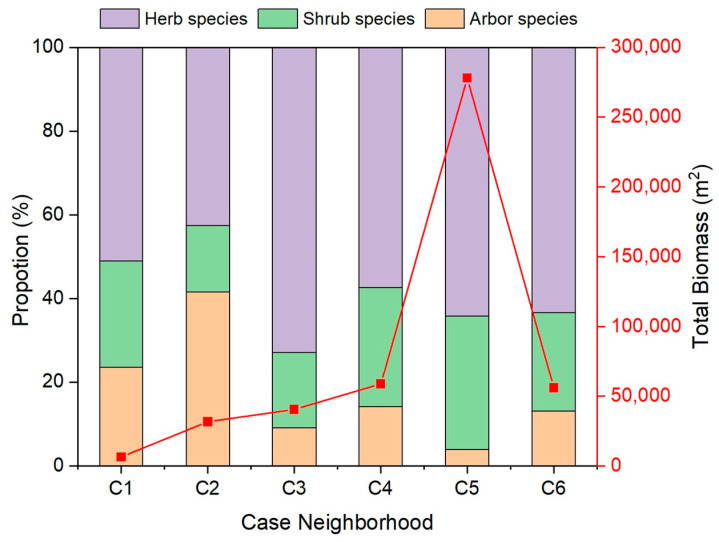
Structure of green quantity of the six case neighborhoods.

**Figure 6 ijerph-19-15242-f006:**
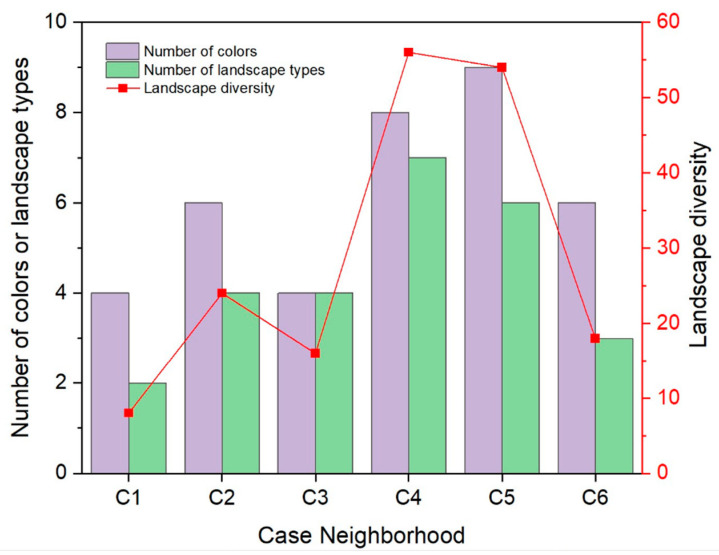
Landscape diversity and its parameters of the six case neighborhoods.

**Figure 7 ijerph-19-15242-f007:**
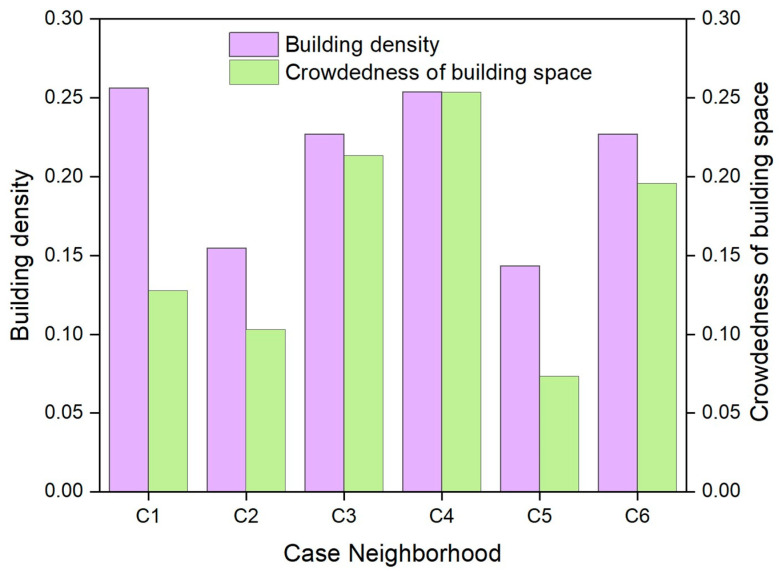
Building density and building space crowdedness of the six case neighborhoods.

**Table 1 ijerph-19-15242-t001:** General situation of neighborhoods.

No.	Block Name	Area (m^2^)	Green Space Rate (%)
C1	West Block of Lian Hua Shang Yuan	17,203.70	15.15
C2	Lian Hua Xin Cheng Zhen Yuan	41,069.60	32.27
C3	East Block of Tian E Mei Yuan	23,726.30	27.37
C4	East Block of Bi Hai Lan Tian	27,079.80	26.34
C5	Shui Jing Hu Jun I	143,747.00	40.44
C6	Quan Shui Wan I	28,524.80	20.95

**Table 2 ijerph-19-15242-t002:** Latent variables and observed variables of SEM.

Latent Variables		Observed Variables		
No.	Name	No.	Name	Description
F1	Physical perception	I1	Visual field	Impacts of building space, light, etc., in landscape space on human visual perception
		I2	Temperature amenity	Comfortable sensation of temperature in residential green space
		I3	Humidity amenity	Comfortable sensation of humidity in residential green space
		I4	Air freshness	Improvement of air freshness by dust removal effect of plants and negative air ions provision
		I5	Plant odor	Scent of trees, grass, and flowers
		I6	Noise	Denoising function provided by plants
F2	Aesthetic cognition	I7	Landscape richness	Landscape and plant component diversity and color richness
		I8	Spatial conformity	Conformity of the space ratio of natural landscapes and buildings
		I9	Management status	The sense of order and the vestige of manual maintenance of ecological infrastructures in the neighborhood
		I10	Green coverage	The area ratio of green space in the neighborhood
F3	Psychological cognition	I11	Safety	The feeling that personal privacy is protected
		I12	Belonging	Senses of identification, affection, and attachment to the neighborhood
		I13	Esteem	Senses of being dignified and respected
		I14	Willingness to communicate	A willingness to stay in public green space where people congregate and to associate with people
F4	Perception about public facilities	I15	Housing quality	Building facilities’ quality
		I16	Transport convenience	Transport condition at the location of residence
		I17	Community service	Services such as cleaning, security, etc.
		I18	Recreation facility	Facilities for fitness, recreation, children’s playground, etc.

**Table 3 ijerph-19-15242-t003:** Vegetation and landscape characteristics, their characterizing indicators, and their corresponding observed variables.

Characteristics of Vegetation and Landscapes	Characterizing Indicators	Observed Variables of Perceptions
Diversity	Shannon–Wiener Index	Landscape richness (I7)
	Complexity of landscape components	
Biomass	Total biomass	Green coverage (I10)
Openness	Building density	Visual field (I1)
	Crowdedness of building space	

**Table 4 ijerph-19-15242-t004:** Structure of green quantity of C1–C6.

	Case Neighborhood	Objective Landscape Characteristic	Residents’ Subjective Perception
Diversity	C1	−0.03	−1.52
	C2	0.04	−0.60
	C3	0.00	−0.20
	C4	−0.61	0.22
	C5	0.94	1.77
	C6	−0.34	0.33
Biomass	C1	−1.66	−1.38
	C2	−1.07	−0.25
	C3	0.33	−0.14
	C4	1.02	0.02
	C5	0.67	1.99
	C6	0.71	−0.24
Landscape openness	C1	−0.25	0.18
	C2	1.07	−0.04
	C3	−0.59	−0.06
	C4	−1.20	−0.27
	C5	1.43	1.81
	C6	−0.45	−1.62

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
