# Peer review of "Resident Satisfaction of Urban Green Spaces through the Lens of Landsenses Ecology"

_ijerph, 2022, doi:10.3390/ijerph192215242_

Round 1

Reviewer 1 Report

Please proof for English language e.g. line 25 posturing should be positing. 

When using an acronym, it should always be spelt out after its first iteration e.g. line 28 what is UGS?

The term Landsenses appears throughout the paper but is never adequately defined. It is not in common usage (in the social sciences) and therefore, its meaning should be clarified.  The fact of the matter is that in the social sciences significant work is carried out across Sociology, Geography and Anthropology that investigates residents' subjective perception of their neighbourhoods, proximate environment and locality.  Many excellent insights have arisen from this work on themes such as sense of attachment to place, sense of belonging, place and placelessness, defensible space, etc.  None of this literature appears here as the focus is very technical and frankly, at one removed from "the residents" even though it purports to focus on subjective cognition and perception. 

Using Maslow's Hierarchy of Needs is quite outdated given the wide body of literature produced in recent years by the disciplines mentioned above. 

Page 4 of 16  

There is a sense in which the paper is setting out to prove technically something that we already know intuitively, and through significant qualitative research in community and neighbourhood studies.  Page 4/line 151: "it turns out that green space quantity and quality are in correlation with residents' satisfaction and happiness."  This is not in the least surprising.  The whole argument here is rather weak as it is difficult to see how the present study either adds value, or new/innovative insight. 

English language formulations on line 199 (page 5 need attention)

The technical element of the paper (data gathering, analysis and exploration seem fine, its just that the findings don't offer much in the way of significant insights. e.g. line 375/6 of course "aesthetic cognition and psychological cognition are directly related to residents' satisfaction." How could it be otherwise given the social science literature that demonstrates the importance of place attachment and belonging?  For me, the paper's weakness is that even though it purports to deal with subjective experience there is no conceptualisation presented of people's social and cultural connectedness to landscape. 

There is also an element of over-determination going on here.  Lines 386-391 suggests a very mechanical relationship between the variables involved, and slightly tendentious conclusions about "improving physical perception can improve psychological cognition, thus improving resident satisfaction." It all feels quite removed from subjective perceptions and responses on the ground. 

I think the conclusions are unsurprising and to some degree, simply state the obvious. A very didactic approach is taken to residents which suggests they have no agency in relation to their perceptions, cognition and tastes.  Lines 400-413. 

Reviewer 2 Report

The authors must be congratulated on working on a critical topic within their discipline. I would like to make some suggestions to further improve this paper:

1) Since the paper focuses on 'residents' satisfaction' and 'perceptions', it is important that the survey respondents are described. While the sample has been presented in a table, it would be useful to know why and how those respondents were contacted - and why were they asked certain demographic questions. Did the researchers expect to see any differences in responses, for example across income levels, etc.?

2) When using structural equation modeling (SEM), it is important to follow its key principles. For example, latent variables are always shown as circles. Similarly, SEM models should be parsimonious. Parsimonious models are simple (minimum number of variables), with optimum predictive power. This does not only apply to the number of factors (variables) in the model but also the number of arrows. I would suggest that the authors try the following model (attached to this review):

Endogenous variable: Resident satisfaction
Mediating variable: Psychological cognition
Exogenous variables: Physical perceptions, Aesthetic cognition, Other Perceptions

Please run covariation (shown with double-headed arrows) between all exogenous variables. 

This suggested model will result in six hypotheses only. Note the fit for this model and the resulting indices. If the fit is good, it is advisable to use the more parsimonious model

3) All SEM models - with latent variables and their relationships - must only be based on theory. For example, if the authors want to demonstrate that two of the exogenous variables may have a direct link to 'residents' satisfaction', then it must be via a relevant theoretical justification. It should also be argued why the third exogenous variable is not expected to result in a direct effect on satisfaction

4) Since the authors are advocating the presence of a mediating variable (i.e. 'psychological cognition') in their model, they must use this term. It is also advised that the authors argue why they expect certain direct effects (e.g., physical perceptions directly impacting satisfaction) and indirect effects (e.g. physical perception indirectly impacting satisfaction via psychological cognition). 

5) I would recommend that the authors find an appropriate concept for 'other perceptions' so that it is theoretically valid and makes a contribution to the literature

Round 2

Reviewer 2 Report

Thank you for undertaking the suggested improvements. My one final suggestion is to change the factor 'scene perceptions' to 'public facilities' or 'perceptions about public facilities'. I will let the Editor decide if this is a useful modification.